# Contribution of *Stenotrophomonas maltophilia* MfsC transporter to protection against diamide and the regulation of its expression by the diamide responsive repressor DitR

**Angkana Boonyakanog**[1☯], **Nisanart Charoenlap**[2,3☯], **Sorayut Chattrakarn**[2], **Paiboon Vattanaviboon**[2,3,4]*, **Skorn Mongkolsuk**[1,2,3]*

**1** Department of Biotechnology, Faculty of Science, Mahidol University, Bangkok, Thailand, **2** Laboratory of Biotechnology, Chulabhorn Research Institute, Bangkok, Thailand, **3** Center of Excellence on Environmental Health and Toxicology (EHT), Bangkok, Thailand, **4** Program in Applied Biological Science: Environmental Health, Chulabhorn Graduate Institute, Chulabhorn Royal Academy, Bangkok, Thailand

☯ These authors contributed equally to this work.
* paiboon@cri.or.th (PV); skorn.mon@mahidol.ac.th (SM)

## Abstract

*Stenotrophomonas maltophilia* contains an operon comprising *mfsB* and *mfsC*, which encode membrane transporters in the major facilitator superfamily (MFS). The results of the topological analysis predicted that both MfsB and MfsC possess 12 transmembrane helices with the N- and C-termini located inside the cells. The deletion of *mfsC* increased the susceptibility to diamide, a chemical oxidizing agent, but not to antibiotics and oxidative stress-generating substances relative to wild-type K279a. Moreover, no altered phenotype was observed against all tested substances for the Δ*mfsB* mutant. The results of the expression analysis revealed that the *mfsBC* expression was significantly induced by exposure to diamide. The diamide-induced gene expression was mediated by DitR, a TetR-type transcriptional regulator encoded by *smlt0547*. A constitutively high expression of *mfsC* in the *ditR* mutant indicated that DitR acts as a transcriptional repressor of *mfsBC* under physiological conditions. Purified DitR was bound to three sites spanning from position + 21 to -57, corresponding to the putative *mfsBC* promoter sequence, thereby interfering with the binding of RNA polymerase. The results of electrophoretic mobility shift assays illustrated that the treatment of purified DitR with diamide caused the release of DitR from the *mfsBC* promoter region, and the diamide sensing mechanism of DitR required two conserved cysteine residues, Cys92 and Cys127. This suggests that exposure to diamide can oxidize DitR through the oxidation of cysteine residues, leading to its release from the promoter, thus allowing *mfsBC* transcription. Overall, MfsC and DitR play a role in adaptive resistance against the diamide of *S. maltophilia*.

**Data Availability Statement:** All relevant data are within the manuscript and its Supporting information files.

**Funding:** This work was supported by Thailand Science Research and Innovation (TSRI), Chulabhorn Research Institute (Grant number 313/2231) and Chulabhorn Graduate Institute (Grant number 2563/007). AB was a recipient of the Science Achievement Scholarship of Thailand from the Office of Higher Education Commission of the Government of Thailand. The funders had no role in study design, data collection and analysis, decision to publish, or preparation of the manuscript.

**Competing interests:** The authors have declared that no competing interests exist.

# Introduction

*Stenotrophomonas maltophilia* is a Gram-negative, rod-shaped, aerobic bacterium, which is ubiquitous in the aqueous environment and soil [1]. It is an opportunistic pathogen causing hospital-acquired infections worldwide, with a relatively high incidence [2]. The treatment of *S. maltophilia* infections is quite challenging because this bacterium is inherently resistant to multiple classes of antibiotics. It deploys several strategies to protect itself from the antimicrobial activity of antibiotics (e.g., the presence of antibiotic efflux transporters and production of antibiotic modifying or degrading enzymes) [3, 4].

Efflux transporters are recognized as an important mechanism responsible for resistance against antibiotics and other toxic compounds [5]. The major facilitator superfamily (MFS) is a membrane transporter superfamily found in all domains of life. MFS members comprise many families that function as secondary active transporters, including uniporters, symporters, and antiporters, which transport substrates across membranes [6]. MFS contributes to transporting a broad spectrum of substrates, including simple monosaccharides, oligosaccharides, drugs, xenobiotics, nucleosides, amino acids, short peptides, lipids, and organic and inorganic anions and cations [6].

The genome of *S. maltophilia* contains genes that encode many transporters [3]; and, those responsible for antibiotic resistance have received much attention due to their multiple antimicrobial resistance characteristics. However, only a few efflux pumps belonging to MFS have been functionally studied, even though many putative *mfs* genes have been annotated in this bacterial species [3, 4]. The MFS tripartite efflux pump EmrCABsm has been shown to be implicated in resistance against tetracycline and carbonyl cyanide *m*-chlorophenyl hydrazone [7], while the MFS SmTcrA is involved in tetracycline resistance [8]. Recently, the contribution of MfsA to resistance against herbicide paraquat and multiple antibiotics, including fluoroquinolones, has been elucidated [9–11]. Therefore, the MFS pump not only plays a role in the antibiotic resistance of *S. maltophilia* but likely increases the ability of bacteria to survive during the saprophytic stage in soil, where *S. maltophilia* is exposed to environmental toxicants, such as pesticides [10].

The expression of several MFS efflux pumps can be induced upon exposure to increased levels of substrates. The *Escherichia coli* TetA-TetR system is one of the most well-characterized MFS gene regulation [12, 13]. TetA is a tetracycline-resistant MFS efflux pump, whose expression is controlled by the TetR transcription repressor. Under normal growth conditions without tetracycline, TetR represses the transcription of *tetA* by binding to the operator region of the *tetA* promoter and negatively affects the ability of RNA polymerase to transcribe *tetA*. Once tetracycline has entered *E. coli*, the drug-$Mg^{2+}$ complex binds the TetR repressor and induces conformational change in TetR, leading to the dissociation of TetR from the *tetA* operator, thus allowing *tetA* transcription. Additionally, transcriptional regulators belonging to other families have been demonstrated to regulate the expression of *mfs* genes. For example, SoxR, a MerR family of transcriptional regulators that senses and responds to a superoxide anion, controls the expression of *S. maltophilia mfsA*, encoding a DHA14 family of MFS [10]. The expression of the *mfs* gene Sm*tcrA* is negatively regulated by SmQnrR, a DeoR family of transcriptional regulators [8]. These adaptive regulations allow the bacteria to withstand harsh environments and sudden changes in environmental conditions.

Here, we are interested in the MFS transporters because of their functional diversity in the extrusion of various substances. In this communication, we report the functional characterization and transcriptional regulation of two MFS encoding genes, *mfsB* (*smlt0548*) and *mfsC* (*smlt0549*). The deletion of *mfsC* renders the mutants increased sensitivity to diamide. The

expression levels of *mfsB* and *mfsC* are controlled by the diamide-inducible regulator *smlt0547*, which we have named DitR.

## Materials and methods

### Bacterial growth conditions

*Stenotrophomonas maltophilia* K279a [3], which is a clinical isolate, was used as a parental wild-type strain. All *S. maltophilia* and *Escherichia coli* strains were aerobically cultured in a lysogeny broth (LB) medium and incubated at 35°C and 37°C, respectively, with continuous shaking at 180 rpm. Routinely, overnight cultures were inoculated into a fresh LB medium at a starting optical density at 600 nm (OD$_{600}$) of 0.1. Exponential-phase cells were used in all experiments.

### Molecular biology techniques

General molecular biology techniques (i.e., genomic DNA and RNA extraction, DNA cloning, agarose gel electrophoresis, Southern blot analysis, polymerase chain reaction (PCR), and *Escherichia coli* transformation) were performed using standard protocols [14]. Plasmid DNA extraction was performed using Qigen plasmid kit (Hilden, Germany). DNA fragments from agarose gel was purified using TIANgel purification kit (TIANGEN, Beijing, China). The transformation of *S. maltophilia* was conducted using electroporation, as previously described [10].

### Construction of Δ*mfsB*, Δ*mfsC*, and Δ*smlt0547* mutants

The unmarked gene deletion mutant of *mfsB* was constructed using the allelic exchange technique with the *cre-lox* antibiotic marker recycling system [15], as previously described [10]. The 809-bp *mfsB* downstream sequence was PCR-amplified using primers BT4198 and BT4199 (Table 1) and K279a genomic DNA as templates prior to cutting with *Apa*I and cloning into the pUC18Gm vector cut with *Apa*I and *Sma*I, yielding pMfsB-GmDown. The 853-bp *mfsB* upstream sequence was amplified using primers BT4196 and BT4197 (Table 1) before being cut with *Eco*RI and cloned into pMfsBGmDown cut with *Nco*I (fill-in with T4 DNA polymerase) and *Eco*RI, producing pMfsB-UpGmDown. This suicide plasmid was transferred into K279a using electroporation. Putative mutants generated from double-crossover events were selected from the gentamicin resistance (45 μg/mL). To excise the Gm$^{r}$ marker, pCM157 [15] expressing Cre-recombinase was introduced into the putative mutant and the resultant unmarked Δ*mfsB* mutants obtained from Cre-mediated *loxP* recombination were screened for the gentamicin sensitive phenotype. The plasmid pCM157 (carrying tetracycline-resistant gene) was cured by subculturing the strains on plates without antibiotic supplementation and screened for the tetracycline-sensitive colonies.

The Δ*mfsC* mutant was constructed using a suicide plasmid pMfsC-UpGmDown. The 946-bp *mfsC* downstream sequence amplified from K279a genomic DNA using primers BT4614 and BT4615 (Table 1) was cut with *Apa*I and cloned into pUC18Gm cut with *Apa*I and *Sma*I. Then, this pMfsC-GmDown plasmid was cut with *Acc*65I (fill-in with Klenow fragment) and *Eco*RI before being ligated with the 951-bp PCR product of the *mfsC* upstream fragment amplified using primers BT4612 and BT4613 (Table 1), which were cut with *Eco*RI. pMfsC-UpGmDown was introduced into K279a, and the unmarked Δ*mfsC* mutant was selected as described for the construction of the Δ*mfsB* mutant.

The Δ*smlt0547* (Δ*ditR*) mutant was constructed using the pSmlt0547-UpGmDown plasmid. The 744-bp *smlt0547* downstream sequence was amplified using primers BT4194 and BT4195

**Table 1. List of primers used in this study.**

| Primers | Sequence (5' → 3') |
|---|---|
| BT4069 | TACCAGAGCGGATGCGGC |
| BT4085 | CGTATGTAAGGAGCTCCG |
| BT4194 | ACGAGGGCCCTGGAAATGCTGTTGC |
| BT4195 | CGGAGGATCCACAACTGCTGACATG |
| BT4196 | GAGCGAATTCGGCAGCATATCCACG |
| BT4197 | CATCGGTACCGGCAGGCGGGC |
| BT4198 | CTGTGGGCCCTACTCGCGCGAC |
| BT4199 | GCCGGATCCCAGGGTCAGCAACG |
| BT4201 | GTTTCTTCCATTTCGGCG |
| BT4202 | GGGGGCGGCGAACAGGTC |
| BT4203 | CATTCTGTGCGTGCTGCT |
| BT4204 | ACCCCCAGAACCGATGCT |
| BT4228 | TCCCCATGGTCCGTCGCACCC |
| BT4229 | GGCTCGAGCCCGGCCGACTTCA |
| BT4250 | TACAAGCCATCCACGCAT |
| BT4334 | CCGACCACCAGCAACAGGG |
| BT4612 | CGCGAATTCGCTGGAGACCGTGTT |
| BT4613 | TGCCCATGGACGCGCCGAGATCTT |
| BT4614 | GTGGGCCCGCATGCGAATCTGTG |
| BT4615 | CAACGCACAGCGCGCCGAGC |
| BT4616 | GCGGCTTTGTCGTTGCAC |
| BT4617 | CGGCAGCATTTCGGTGGT |
| BT4669 | AGCGCGCCCGTGCCTATCTGGAG |
| BT4670 | AGGCACGGG CGCGCTGTACGAAA |
| BT4671 | CGCCACGCCGTCCACTCGATGCA |
| BT4672 | GTGGACGGCGTGGCGCTGCGATT |
| BT4686 | CTGCATCCACCTCCCCTG |
| M13 Forward | GTAAAACGACGGCCAGTG |
| M13 Reverse | AACAGCTATGACCATGAT |
| Oligo(dT) | CGTATCGATGTCGACTTTTTTTTTTTTTTTT |

(Table 1) and K279a genomic DNA as templates. The PCR product was cut with *Apa*I and cloned into pUC18Gm cut with *Apa*I and *Sma*I. Then, this pSmlt0547-GmDown plasmid was cut with *Eco*RI (fill-in with Klenow fragment) before being ligated with the 716-bp PCR product of the *mfsC* upstream fragment amplified using primers BT4334 and BT4201 (Table 1). Then, pSmlt0547-UpGmDown was introduced into K279a, and the unmarked Δ*ditR* mutant was selected as described for the construction of Δ*mfsB* mutant. All Δ*mfsB*, Δ*mfsC*, and Δ*ditR* mutants were confirmed using PCR with primers flanking the deletion sequence and verified by Southern blot analysis.

## Construction of *mfsC* and *ditR* expression plasmids

The *mfsC* full-length gene was amplified from K279a genomic DNA using primers BT4686 and BT4250 (Table 1). The 1231-bp PCR product was cloned into pBBR1MCS [16] at the *Sma*I site, yielding pMfsC for the *mfsC* expression under the *lacZ* promoter. pSmlt0547 or pDitR for the *ditR* expression was constructed by cloning 654-bp PCR products containing

*ditR* full-length gene amplified using primers BT4085 and BT4069 (Table 1) into pBBR1MCS cut with *Sma*I.

## Amino acid sequence alignment, phylogenetic and membrane topology analyses

The amino acid sequence alignment was performed using Clustal omega algorithm [17]. Phylogenetic tree was constructed by the neighbor-joining method with 500 bootstrap replicates using MEGA 11.0 software [18]. Protein topology of MfsB and MfsC was analyzed using the PROTTER algorithm [19].

## Site-directed mutagenesis of DitR

Site-directed mutagenesis was performed to convert Cys92 and Cys127 of DitR to alanine using a two-step, PCR-based method, as previously described [20]. pDitR was used as a template for amplification with a mutagenized primer pair and the vector primers M13-forward and M13-reverse. The mutagenized primer pairs used were BT4669 and BT4670 (Table 1) for DitR$_{C92A}$ and BT4671 and BT4672 (Table 1) for DitR$_{C127A}$. After the two-step PCR, the products were digested with *Kpn*I and *Xho*I and cloned into pBBR1MCS cut with the same enzymes, generating pDitR$_{C92A}$ and pDitR$_{C127A}$. The nucleotide sequences of the DNA insert were determined to confirm that no unexpected mutation had occurred. pDitR$_{C92,127A}$ was constructed as described for pDitR$_{C127A}$, except for pDitR$_{C92A}$, which was used as the DNA template.

## Construction of *mfsBC* promoter-*lacZ* fusion

pP$_{mfsBC}$-lacZ, carrying the *mfsBC* promoter transcriptionally fused to the *lacZ* reporter gene, was constructed. The *mfsBC* promoter sequence was amplified using primers BT4201 and BT4202 (Table 1) and K279a genomic DNA as templates, and the 204-bp PCR product was cloned into the *Sma*I-cut pUFR027*lacZ* plasmid [21] containing a promoterless *lacZ*, yielding pP$_{mfsBC}$-lacZ.

## Determination of resistance levels to oxidants

A plate sensitivity assay was performed to determine the oxidant resistance levels, as previously described, with some modifications [10]. The exponential-phase cultures of *S. maltophilia* K279a and the mutant strains were 10-fold serially diluted in a fresh LB broth, and 10 μL of each dilution was spotted onto an LB plate containing various oxidants (750 μM diamide, 75 μM N-ethyl maleimide [NEM], 200 μM H$_2$O$_2$, 75 μM cumene hydroperoxide, 200 μM plumbagin or 500 μM menadione). The number of colony forming units (CFUs) growing on the plates was scored after incubation at 35˚C overnight. The resistance level against the oxidants was expressed as the percent survivals, defined as the CFU percentage on plates containing oxidants over that on the control plates without oxidants.

## Antibiotic susceptibility testing

Antibiotic susceptibility on *S. maltophilia* K279a and mutant strains was determined using standard Kirby-Bauer disc diffusion assay [22]. The antibiotic discs, including carbenicillin (CAR100), cefepime (FEP30), ceftazidime (CAZ30), amikacin (AK30), gentamicin (CN30), netilmicin (NET30), neomycin (N30), levofloxacin (LEV5), moxifloxacin (MXF5), ofloxacin (OFX5), chloramphenicol (C30), tetracycline (TE30), tigecycline (TGC15), sulbactam/cefoperazone (SCF105), amoxicillin/clavulanic acid (AMC30), sulfamethoxazole/ trimethoprim

(SXT25), polymyxin B (PB300), fosfomycin (FOS50) were purchased from Thermo Fisher Oxoid (Basingstoke, UK). Discs containing benzalkonium chloride (BAC, 0.5 mg) and acriflavine (ACR, 0.5 mg) were prepared using 6-mm-diameter discs soaked with 5 μL of 100 mg/mL BAC and ACR, respectively.

## Real-time reverse transcriptase PCR

Expression analysis using real-time reverse transcriptase PCR (RT-PCR) was performed as previously described [10]. Exponential-phase *S. maltophilia* strains were challenged with the tested substances at sub-minimum inhibition concentrations i.e.100 μM $H_2O_2$, 50 μM cumene hydroperoxide, 50 μM NEM, 200 μM menadione, 100 μM plumbagin, 100 μM paraquat, 500 μM diamide, 32 μg/ml acriflavine, 200 μM phenazine, and 8 μg/ml benzalkonium chloride for 30 min before the cells were harvested for total RNA extraction. Reverse transcription reaction was performed using 5 μg of DNase I-treated RNA, random hexamers, and the RevertAid M-MuLV reverse transcriptase kit (Fermentas, Latvia). RT-PCR was performed using 20 ng of cDNA, an *mfsC* specific primer pair (BT4203 and BT4204 [Table 1]), and SYBR green PCR master mix (Applied Biosystems, Thermo Fisher Scientific, USA). The 16S ribosomal RNA gene amplified using primers BT2781 and BT2782 (Table 1) was used as the normalizing gene. The PCRs were run on an Applied Biosystems StepOne Plus for 40 cycles under the following conditions: denaturation at 95˚C for 30 s, annealing at 58˚C for 45 s, and extension at 72˚C for 45 s. The relative expression was calculated using StepOne software v2.1 and expressed as fold expression over the uninduced level of K279a. For each experiment, at least three independent biological repeats were performed.

## Purification of DitR protein

The 6×His-tag DitR protein was highly expressed using the pET-blue2 expression vector and *E. coli* system (Novagen). The 598-bp PCR product containing full-length *ditR*, amplified using primers BT4228 and BT4229 (Table 1) and K279a genomic DNA as templates, was cloned into pETBlue-2 at *Nco*I and *Xho*I sites, yielding pET-DitR. To purify DitR, an exponential-phase culture of *E. coli* BL21 (DE3)/pLacI harboring pET-DitR was induced with 250 μM isopropyl-β-D-thiogalactopyranoside overnight at 25˚C. The 6×His-tag DitR protein was purified as previously described [23] using a column containing nickel-nitrilotriacetic acid resin (Qiagen, France) and elute with 150 mM imidazole. The purity of purified DitR was estimated by SDS-PAGE and densitometric analysis of the Coomassie blue-stained gels (S1 Fig). Fractions with DitR purity greater than 95% were used in further experiments.

## Electrophoretic mobility shift assay

The putative *mfsBC* promoter sequence (204-bp), amplified using primers [32]P-labeled BT4201 and BT4202 (Table 1) and pP*mfsBC*-lacZ as DNA templates, was used as a probe. Electrophoretic mobility shift assay (EMSA) was conducted as previously described [10] using increasing amounts of purified DitR protein (0–100 nM) and 20 ng of [32]P-labeled probe in the reaction mixture. The reaction was incubated at room temperature for 20 min. The protein–DNA complexes were analyzed on a 7% non-denaturing polyacrylamide gel electrophoresis (PAGE).

The effects of diamide and dithiothreitol (DTT) on the DNA binding activity of purified DitR were determined by EMSA. Binding reactions containing 50 nM purified DitR protein and 20 ng of [32]P-labeled probe were treated 0.25, 0.5 and 1 mM diamide at room temperature for 5 min prior to PAGE. The effects of DTT on the DNA binding activity of the diamide-treated DitR were determined by treating the diamide (1 mM)-treated DitR with 0, 1 and 5 mM DTT at room temperature for 5 min.

### DNase I footprinting assay

The DNase I protection assay was performed as previously described [23]. Briefly, the binding reaction was set in a 20-μL reaction mixture containing 50 ng [$^{32}$P]-labeled 204-bp *mfsBC* promoter fragments prepared as described for EMSA and increasing concentrations of purified DitR (0–50 nM). After incubation for 20 min, the binding mixture was treated with 0.25 unit DNase I for 30 s. The digested DNA was separated on a 6% polyacrylamide urea denaturing gel alongside the sequencing ladder.

### 5′ rapid amplification of cDNA ends

The 5′ rapid amplification of cDNA ends (RACE) was performed using a 5′/3′ RACE kit (Roche, Germany), as previously described [10]. The DNase I-treated total RNA was reverse-transcribed using a specific primer BT4204 (SP1). The first-strand cDNA was purified prior to adding poly(A) to the 5′-terminus using terminal transferase enzyme. The poly(A)-tailed cDNA was then PCR-amplified using a specific primer BT4617 (SP2) and an anchored oligo (dT) primer. The PCR product was cloned into pUC18, and the transcription start site (+1) was identified by DNA sequencing.

### Enzymatic assay

Crude bacterial cell lysates were prepared from exponential-phase cultures, and protein assays were performed as previously described [20]. The β-galactosidase activity was determined using *o*-nitrophenyl-β-D-galactoside as a substrate, as previously described, and expressed as an international unit defined as the amount of enzyme generating 1 μmol of *o*-nitrophenol per min at 25˚C [24]. Data shown are the mean ± SD of triplicate experiments.

## Results and discussion

The contribution of MFS efflux pumps to the resistance of *S. maltophilia* against herbicide paraquat, antibiotics, and disinfectant quaternary ammonium compounds (QACs) has been previously reported [9, 10, 25]. The genome of *S. maltophilia* contains many putative genes encoding MFS with an unknown function [3, 26]. Here, we reported the functional and regulation characterization of *smlt0548* and *smlt0549* encoding two putative MFS transporters (hereafter referred to as *mfsB* and *mfsC*, respectively). This gene organization is conserved in all *Stenotrophomonas* strains, whose genome sequences are available in the Kyoto Encyclopedia of Genes and Genomes (KEGG) database.

### MfsB and MfsC belong to the 12 transmembrane domain subfamily of MFS

The *mfsB* and *mfsC* encode 404- and 379-amino-acid proteins, respectively. Prokaryotic MFS transporters can be divided into two main clusters: the drug:H$^+$ antiporter with 12 transmembrane α-helical domain (DHA12) and those with 14 transmembrane α-helical domain (DHA14) [27, 28]. The results from the membrane topology analysis of the deduced amino acid sequences of *S. maltophilia* MfsB and MfsC using the PROTTER algorithm [19] suggested that they are efflux pumps that belong to the DHA12 family. MfsB shares a low identity (less than 20%) with known and characterized MFS. MfsC shares 28.8% identity, which is the highest, with *E. coli* YicM (NepI), an efflux pump of purine ribonucleosides [29]. The results from phylogenetic analysis also showed that MfsC is closely related to YicM (Fig 1). Moreover, the fact that MfsB and MfsC share only 24.8% identity for the amino acid sequences and 29.7% identity for the nucleotide sequences, together with the data from phylogenetic tree analysis,

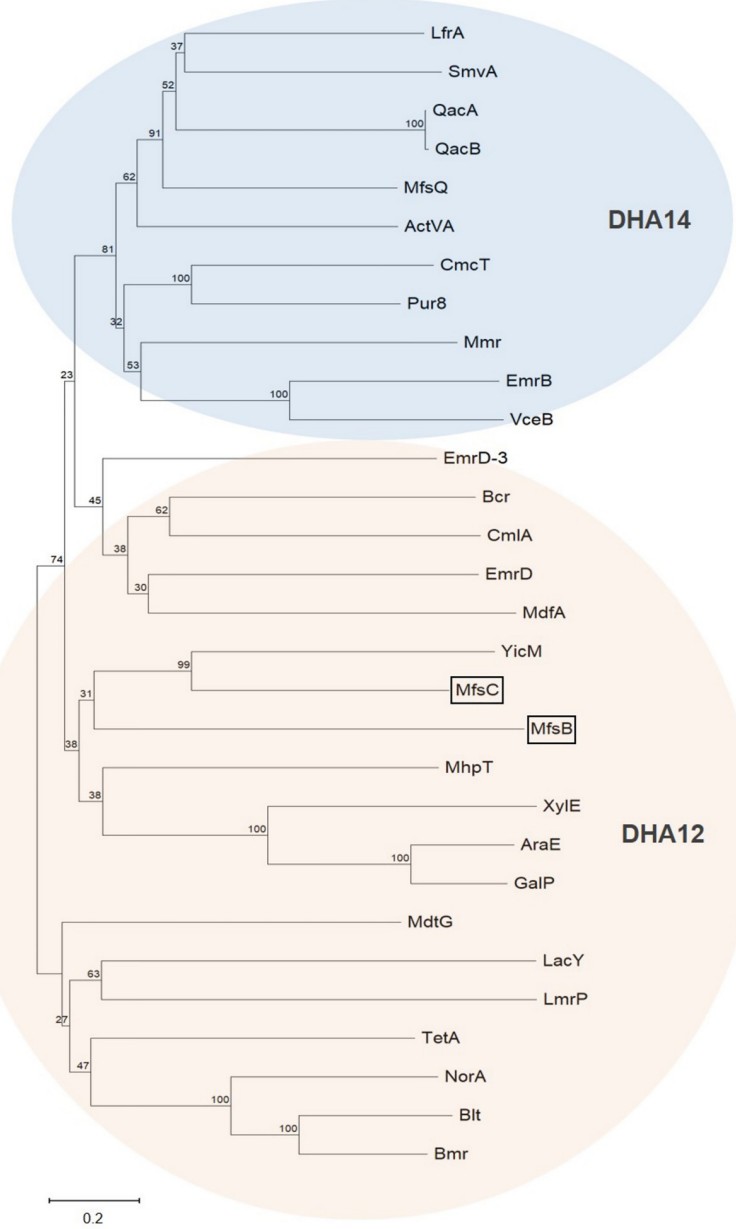

**Fig 1. Phylogenetic analysis of *S. maltophilia* MfsC.** Phylogenetic tree was constructed by the neighbor-joining method with 500 bootstrap replicates using MEGA11 software. Amino acid sequences of known MFS belonging to DHA14 and DHA12 from various bacteria were retrieved from the UniProt (https://www.uniprot.org). DHA14; ActVA (Q53903), CmcT (Q04733), EmrB (P0AEJ0), LfrA (A0R5K5), MfsQ (B2FTN0), Mmr (P11545), Pur8 (P42670), QacA (Q1XG09), QacB (Q7WUJ5), SmvA (P37594), VceB (O51919). DHA12; AraE (P0AE24), Bcr (P28246), Blt (P39843), Bmr (P33449), CmlA (Q83V15), EmrD (P31442), EmrD-3 (C3LUT7), GalP (P0AEP1), LacY (P02920), LmrP (Q48658), MdfA (P0AEY8), MdtG (P25744), MhpT (P77589), NorA (P0A0J7), TetA (P02981), XylE (P0AGF4), YicM (J7R7Q1).

suggests that the presence of *mfsB* and *mfsC* gene clusters is not attributable to gene duplication.

## *mfsC* mutant is susceptible to diamide

To evaluate the physiological function of *mfsB* and *mfsC* in *S. maltophilia*, the Δ*mfsB* and Δ*mfsC* mutants were constructed and their phenotypes against various stress-producing substances were determined. The phenotypic antibiotic susceptibility of the mutants was initially determined using a standard Kirby-Bauer disc diffusion method [22] with various antibiotics. No alterations in the antibiotic susceptibility profiles of the Δ*mfsB* and Δ*mfsC* mutants relative to the wild-type K279a were observed for all antibiotics tested, including those belonging to β-lactams, fluoroquinolones, amino glycosides, and macrolides (Fig 2A). It is likely that MsfB and MfsC play no role in the antibiotic resistance of *S. maltophilia*.

(A)

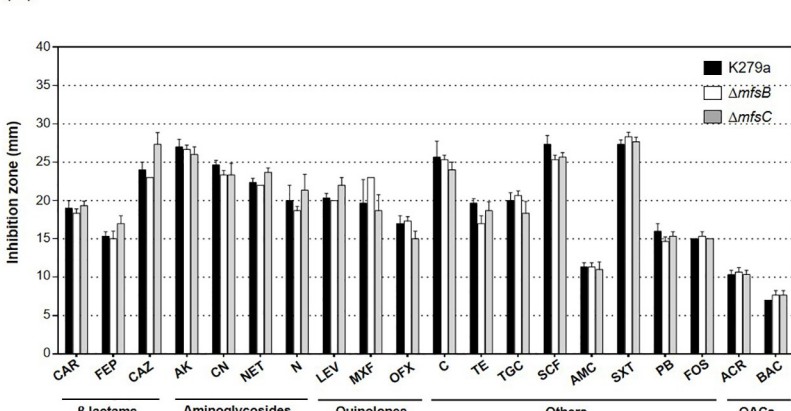

(B)

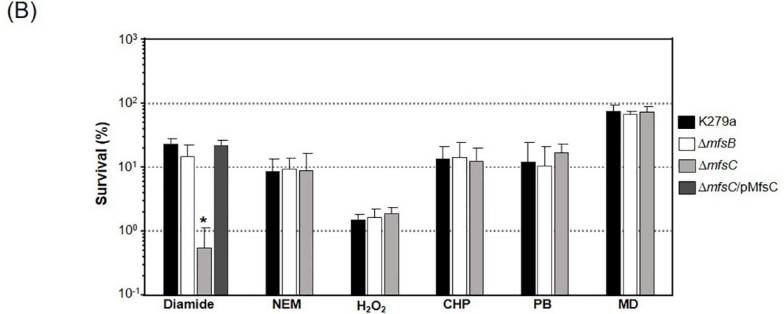

**Fig 2. Phenotypic analysis of Δ*mfsB* and Δ*mfsC* mutant strains.** (A) Antibiotic susceptibility of wild-type K279a (closed bar), Δ*mfsB* (open bar) and Δ*mfsC* (light-gray bar) mutant strains was determined using standard Kirby-Bauer disc diffusion assay [22]. The representative antibiotics from various classes including carbenicillin (CAR), cefepime (FEP), ceftazidime (CAZ), amikacin (AK), gentamicin (CN), netilmicin (NET), neomycin (N), levofloxacin (LEV), moxifloxacin (MXF), ofloxacin (OFX), chloramphenicol (C), tetracycline (TE), tigecycline (TGC), sulbactam/cefoperazone (SCF), amoxycilin/clavulanic acid (AMC), sulfamethoxazole/ trimethoprim (SXT), polymyxin B (PB), fosfomycin (FOS) were tested. Benzalkonium chloride (BAC) and acriflavine (ACR) were the representative of quaternary ammonium compound disinfectants (QACs). (B) The resistance levels of *S. maltophilia* K279a (closed bar), Δ*mfsB* (open bar), Δ*mfsC* (light-gray bar), and Δ*mfsC* harboring pMfsC (Δ*mfsC*/pMfsC, dark-gray bar) against the oxidants were determined using a plate sensitivity assay. Exponential cells were plated onto plates containing 750 μM diamide, 75 μM NEM, 250 μM $H_2O_2$, 75 μM cumene hydroperoxide (CHP), 200 μM plumbagin (PB), and 500 μM menadione (MD). Asterisk (*) indicates significant differences compared to K279a according to Dunnett's post-hoc test (p-value < 0.05).

Next, the mutant phenotypes against ROS-generating agents and oxidants were determined using plate sensitivity assays [10]. As illustrated in Fig 2B, the Δ*mfsC* mutant showed roughly 10-fold more susceptibility to diamide, a thiol-depleting agent, than the wild-type K279a. A plasmid-borne expression of *msfC* from the pMfsC plasmid, which is a medium-copy-number expression vector pBBR1MCS carrying full-length *mfsC*, fully complements the diamide-susceptible phenotype of the Δ*mfsC* mutant (Fig 2B). The Δ*mfsC* mutant showed no altered susceptibility levels to NEM, a thiol-depleting electrophile, and other oxidants, including $H_2O_2$, organic hydroperoxide (cumene hydroperoxide), and superoxide generators/redox cycling drugs (menadione and plumbagin) relative to K279a (Fig 2B). The Δ*mfsB* mutant showed no phenotypic changes to all oxidants tested (Fig 2B). As the Δ*mfsC* mutant showed a susceptible phonotype to diamide but not to other ROS-generating agents and NEM, we speculated that the diamide-susceptible phenotype of the mutant is not attributable to the defects in the oxidative stress response. Thus, MfsC might be associated with diamide efflux transport. However, we cannot rule out the involvement of MfsC in the transportation of substances required for the protection of *S. maltophilia* from diamide toxicity.

## Expression of *mfsC* is induced by diamide

The expression levels of *mfsC* in response to stresses, including oxidative stress and disinfectants, were determined using RT-PCR. Total RNA was extracted from exponential-phase cultures of *S. maltophilia* wild-type K279a grown under uninduced conditions and induced with sublethal concentrations of diamide, $H_2O_2$, cumene hydroperoxide, electrophile NEM, redox cycling drugs (menadione, plumbagin, paraquat, and phenazine), and biocide QACs (acriflavine and benzalkonium chloride) [10, 25]. The 16S ribosomal RNA was used as the normalizing gene. As shown in Fig 3, among all treated chemicals, diamide was the most potent inducer, as it could induce 16.3 ± 4.0-fold *mfsC* expression relative to the uninduced control. The treatment with benzalkonium chloride and NEM slightly induced *mfsC* expression by 1.7 ± 0.1-fold and 1.5 ± 0.3-fold, respectively, while other tested substances failed to induce *mfsC* expression (Fig 3).

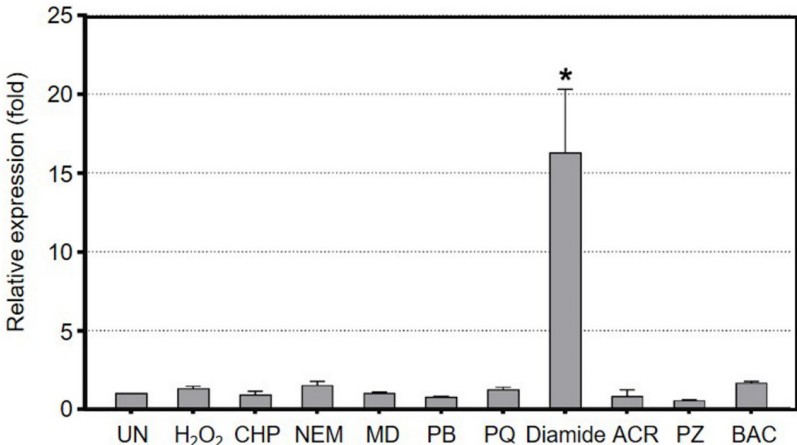

**Fig 3. Expression profile of *mfsC* in K279a.** RT-PCR was conducted to determine the expression level of *mfsC* in *S. maltophilia* K279a. Total RNA was extracted from the cultures grown under uninduced (UN) condition and induced with 100 μM $H_2O_2$, 50 μM cumene hydroperoxide (CHP), 50 μM NEM, 200 μM menadione (MD), 100 μM plumbagin (PB), 100 μM paraquat (PQ), 500 μM diamide, 32 μg/ml acriflavine (ACR), 200 μM phenazine (PZ), and 8 μg/ml benzalkonium chloride (BAC). The experiments were conducted in triplicate, and data are presented as mean ± SD. Asterisk (*) indicates significant differences compared to the uninduced level according to the Dunnett's post-hoc test (p-value < 0.05).

(A)

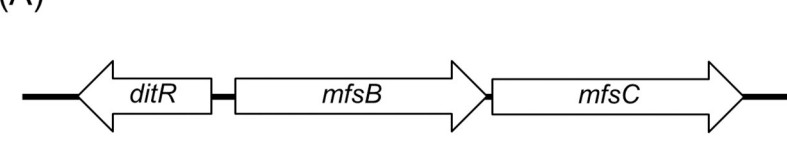

(B)

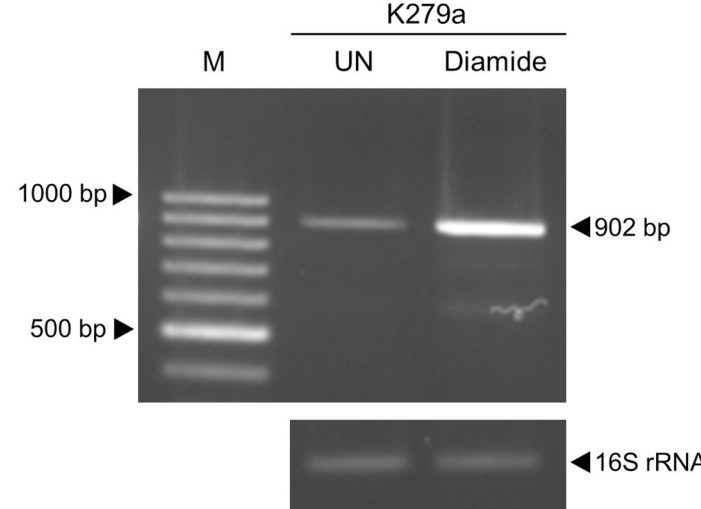

**Fig 4. Gene organization and operon structure analysis of *mfsB* and *mfsC*.** (A) Gene organization of *ditR*, *mfsB* and *mfsC* in *S. maltophilia* K279a. (B) Operon structure analysis of *mfsB* and *mfsC* was performed using RT-PCR. Total RNA were extracted from *S. maltophilia* K279a cultures grown under uninduced condition (UN) and induction with 500 μM diamide. After reverse transcription, cDNA was used as templates for PCR amplification using primers BT4612 and BT4617 flanking intergenic region of *mfsB* and *mfsC* with an expected size for the PCR product of 902 bp. The 16S ribosomal RNA was used as a loading control. M represents 100-bp DNA ladder.

## *mfsB* and *mfsC* are transcribed as an operon

As *mfsB* and *mfsC* are separated by 29 bp, they represent an operon structure (Fig 4A). To prove this speculation, end-point RT-PCR experiments using the cDNA of *S. maltophilia* wild-type K279a cultivated under the induced conditions and primers flanking an intergenic region between *mfsB* and *mfsC* were conducted. A PCR product of approximately 900 bp, corresponding to the expected size, was obtained (Fig 4B). The results indicated that *mfsB* and *mfsC* were transcribed as an operon and the expression of *mfsB* and *mfsC* could be induced by diamide treatments (Fig 4B). Although *mfsB* was co-transcribed with *mfsC* in an diamide-inducible manner, it seems likely that MfsB has no role in the protection of *S. maltophilia* against all tested antibiotics and oxidants (Fig 2). The precise physiological role of *S. maltophilia* MfsB is as yet unknown and required further investigation.

## Smlt0547 regulates *mfsBC* expression as a transcriptional repressor

According to K279a genome, *smlt0547*, encoding a transcriptional regulator in the TetR/AcrR family, is located immediately upstream of the *mfsBC* operon in a divergent orientation. This gene arrangement suggests that Smlt0547 can regulate the expression of the *mfsBC* operon.

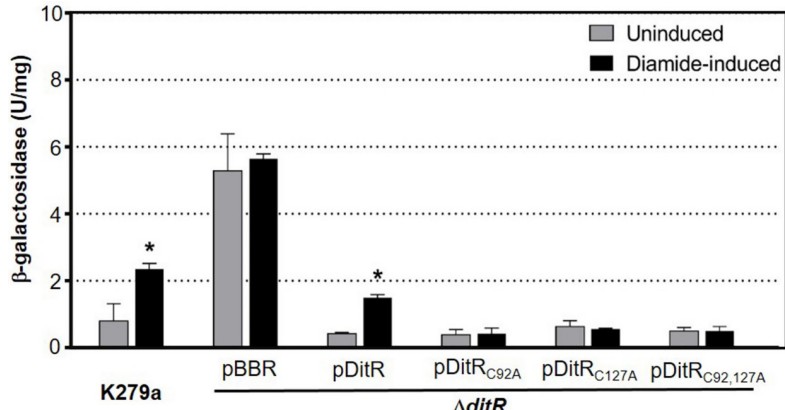

**Fig 5. *mfsBC* promoter activity in *S. maltophilia* strains.** The plasmid pP*mfsBC*-lacZ containing *mfsBC* promoter-*lacZ* fusion was introduced into *S. maltophilia* K279 and Δ*ditR* mutant, which harbored the pBBR1MCS vector (Δ*ditR*/pBBR), pDitR (Δ*ditR*/pDitR), pDitR$_{C92A}$ (Δ*ditR*/pDitR$_{C92A}$), pDitR$_{C127A}$ (Δ*ditR*/pDitR$_{C127A}$), or pDitR$_{C92,127A}$ (Δ*ditR*/pDitR$_{C92,127A}$). Each strain was cultivated to an exponential phase before being induced with 500 μM diamide. The β-galactosidase activity in the clear lysates was determined. Experiments were performed in triplicate, and data are presented as mean ± SD. Asterisk (*) indicates significant differences of the diamide-induced versus uninduced (UN) condition in each strain using a paired *t*-test (p-value < 0.05).

This gene organization is found in all *Stenotrophomonas* genome sequences available in the KEGG database [3, 26]. Certainly, this gene cluster also exists in diverse bacteria.

To assess the role of Smlt0547 in the regulation of the *mfsBC* operon, the deletion mutant of *smlt0547* was constructed, and the expression profile in response to the stresses of *mfsC* was examined using *mfsBC* promoter-*lacZ* fusion. The Δ*smlt0547* mutant harboring pP*mfsBC*-lacZ was grown to an exponential phase and challenged with various substances. The β-galactosidase activity from cell clear lysates was determined. The *mfsBC* promoter activity in uninduced cultures of the Δ*smlt0547* mutant was comparatively higher (5.3 ± 1.0) than that of wild-type K279a (0.8 ± 0.5), and a constitutively high promoter activity was observed for treatment with diamide (5.7 ± 0.2) (Fig 5). This result indicates that Smlt0547 functions as a repressor regulating *mfsBC* expression in a diamide-inducible manner, and hereafter Smlt0547 is referred to as a diamide-inducible transcriptional repressor, DitR.

The alignment of DitR-deduced amino acid sequence with other known transcriptional regulators belonging to the TetR family was performed and revealed that DitR shares 28.7% identity, which is the highest score, with TtgW from *Pseudomonas putida* and 25.9% identity with its paralogous SmeT. TtgW is a putative transcriptional regulator with an unknown function; it is expressed as an operon with TtgV, a transcriptional repressor of the *ttgGHI* operon encoding the solvent efflux pump [30]. SmeT acts as a transcriptional repressor of the multidrug efflux pump SmeDEF [31]. The potent inducers of both TtgW and SmeT are unknown. The binding of biocide triclosan to SmeT, however, can derepress the *smeDEF* expression [32]. Phylogenetic tree analysis also showed that DitR was evolutionarily close to *P. putida* TtgW (Fig 6).

## Purified DitR specifically binds to the *mfsBC* promoter region

The transcriptional regulator in the TetR family commonly exerts its regulatory function by binding to the target promoter. To evaluate whether the DitR protein binds to the *mfsBC* promoter region, the 6×His-tag DitR protein was purified and an EMSA assay was performed. As illustrated in Fig 7A, the shifted band, which is a purified DitR protein and an *mfsBC* promoter complex, was observed at a concentration of 1 nM. When the concentration of DitR protein

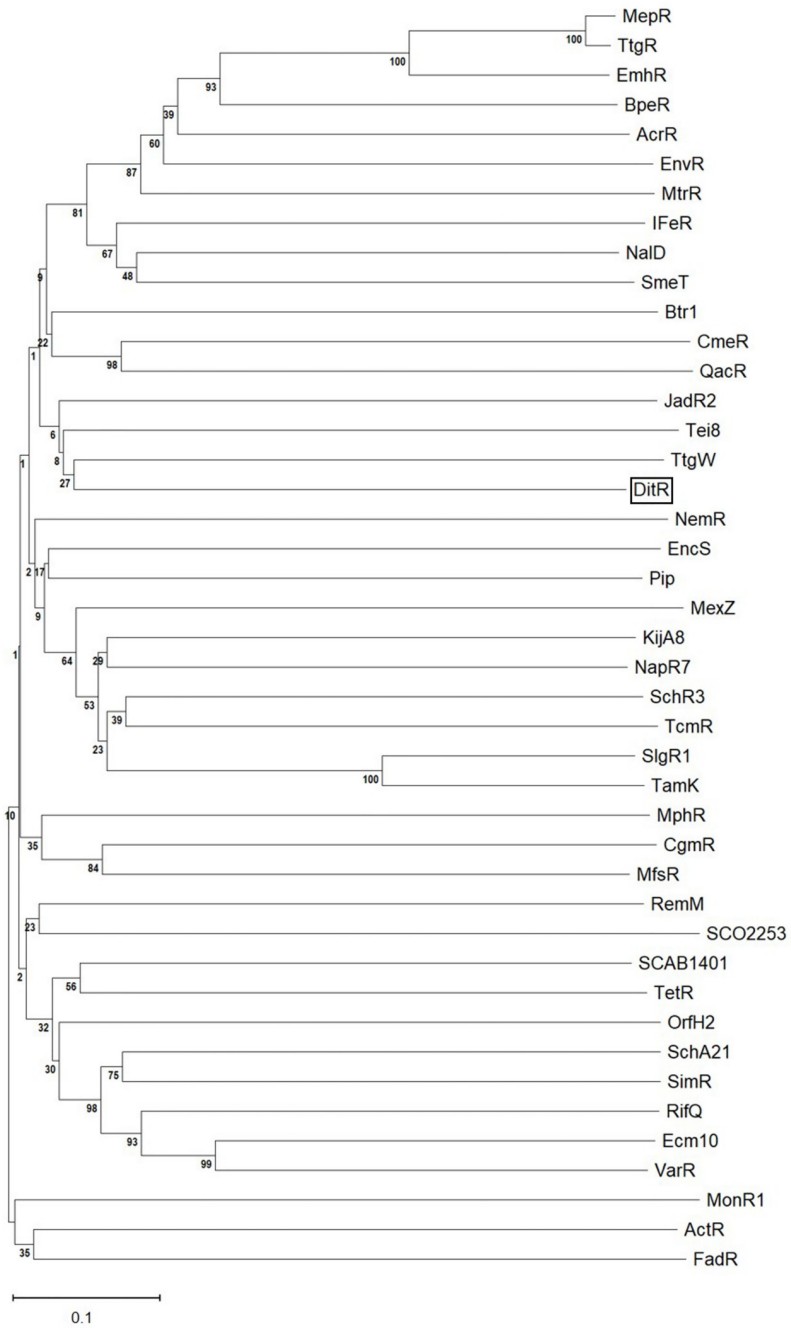

**Fig 6. Phylogenetic analysis of *S.maltophilia* DitR.** Phylogenetic analysis was constructed by Molecular Evolutionary Genetics Analysis (MEGA) version 11 using neighbor-joining method with 500 bootstraps. Amino acid sequences of known transcription regulators belonging TetR family from various bacteria were retrieved from the NCBI databases; AcrR (NP_414997.1), ActR (WP_012067478.1), BpeR (WP_004526226.1), Btr1 (BAE07057.1), CgmR (CGL2612 [CAF21274.1]), CmeR (WP_002843095.1), Ecm10 (BAE98159.1), EmhR (WP_008080104.1), EncS (AAF81739.1), EnvR (EGT68974.1), FadR (NP_415705.1), IFeR (AAC25692.1), JadR2 (AAB36583.1), KijA8 (ACB46470.1), MepR (BAF76382.1), MexZ (QGX60202.1), MfsR (CAQ46150.1), MonR1 (AAO65809.1), MphR (AAS13766.1), MtrR (P39897.1), NalD (NP_252264.1), NapR7 (ABS50469.1), NemR (NP_416166.3), OrfH2 (BAC68287), Pip (ACO79951.1), QacR (BAE92852.1), RemM (CAE51183.1), RifQ (GCB89472.1), SCAB1401 (CBG67396.1), SchA21 (CAH10121.1), SchR3 (AEH42489.1), SCO2253 (NP_626502.1), SimR (3ZQL_A), SlgR1 (CBA11576.1), TamK (ADC79649.1), TcmR (AIS01202.1), Tei8 (CAE53340.1), TetR (WP_000470728.1), TtgR (RNF87112.1), TtgW (VCT90291.1), and VarR (BAB32408.1).

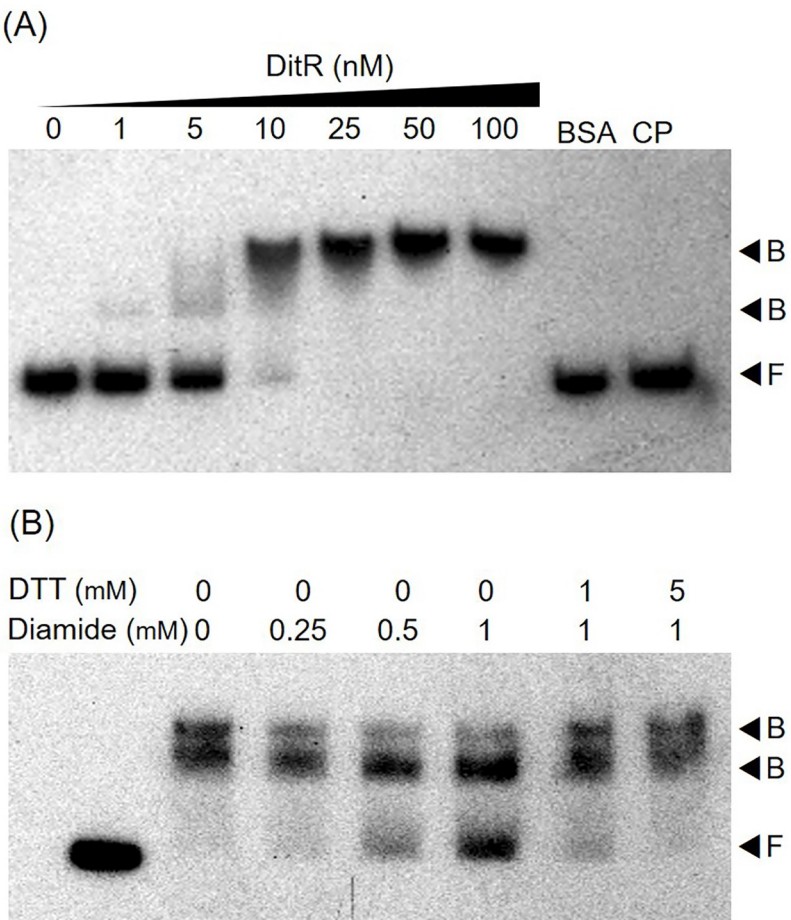

**Fig 7. Binding of purified DitR to *mfsBC* promoter.** (A) EMSA was performed using increased amounts of purified DitR (0–100 nM) and a $^{32}$P-labeled 204-bp *mfsBC* promoter region. BSA and CP signify unrelated protein bovine serum albumin (500 μM) and cold probe, which is an unlabeled *mfsBC* promoter fragment (100 ng). (B) EMSA experiments were repeated using 50 nM of DitR. After the binding reaction was completed, the mixtures were treated with the indicated concentrations of diamide (0–1 mM). In the DTT-treated experiments, the binding reaction was treated with 1 mM diamide prior to being treated with 1 and 5 mM DTT. Free (F) and bound (B) probes are marked by arrowheads.

was gradually increased, two shifted bands were observed, and a completed shift was obtained at the DitR concentration of 25 nM. The presence of two different band shifts of the protein-DNA complexes might have resulted from the binding of DitR occurring at multiple protein binding sites on the *mfsBC* promoter sequence. No binding complexes were observed when excess unrelated protein, bovine serum albumin (BSA), was added to the binding reaction instead of DitR protein, and excess unlabeled probe (CP) could compete with the labeled probe in the DitR-binding complexes, indicating the specificity of DitR binding to the *ditR-mfsBC* promoter region (Fig 7A). The ability of DitR to bind to the *mfsBC* promoter sequence indicated that the *mfsBC* expression was directly regulated by DitR.

## Treating with diamide releases DitR from DitR-*mfsBC* promoter binding complexes

The result of the *in vivo* experiments showed that the *mfsC* expression could be induced in response to diamide exposure. We hypothesized that, as a member of the TetR family, DitR

binds to the *msfBC* promoter and represses its transcription. The presence of diamide causes the release of DitR from the target promoter. Thus, we determined whether diamide interacts with purified DitR. EMSA experiments were repeated using 50 nM purified DitR. After the incubation, the binding reaction was treated with various concentrations of diamide before electrophoresis. As shown in Fig 7B, the free probe band was gradually increased upon treatment with enhanced concentrations of diamide (0.25–1.0 mM). The result suggested that diamide can react with purified DitR protein, leading to the release of DitR protein from the *mfsBC* promoter. Diamide behaves as an oxidizing agent capable of reacting with thiol groups in protein. We then postulated that DitR is oxidized by diamide, causing a conformational change in the protein structure and reducing the binding affinity to the *mfsBC* promoter. To confirm this speculation, the reducing agent dithiothreitol (DTT) was used in the diamide-treated EMSA experiment to reduce the diamide-oxidized DitR protein. DTT at concentrations of 1 and 5 mM was added to the binding reactions treated with 1 mM diamide. The results showed that, after treatment with DTT, the bound probe (DitR-*mfsBC* promoter complex) steadily increased in a DTT concentration-dependent manner (Fig 7B). This result indicates that the purified DitR protein can be reversibly oxidized by diamide, leading to the release of DitR protein from the *mfsBC* promoter region.

The amino acid sequence alignment of *S. maltophilia* DitR with the deduced amino acid sequences derived from its orthologous coding DNA sequences, retrieved from the KEGG database [26], was conducted using the Clustal omega algorithm [17]. The DitR orthologs exist in various bacterial species, including *Erwinia persicina*, *Yersinia canariae*, *Enterobacter sp*, *Pectobacterium carotovorum*, *Xanthomonas arboricola*, *Serratia fonticola*, *Pseudomonas cremoricolorata*, *Pseudomonas putida*, *Alcaligenes faecalis*, and *Acinetobacter bereziniae*. The alignment result revealed two conserved cysteine residues, Cys92 and Cys127 (Fig 8), which were speculated to be targets of diamide oxidation. Site-directed mutagenesis was performed to assess the role of Cys92 and Cys127 in diamide sensing of DitR by replacing either Cys92 or Cys127 with Ala in pDitR expression plasmid, yielding pDitR$_{C92A}$, pDitR$_{C127A}$, and pDitR$_{C92,127A}$. The plasmids were then introduced into the Δ*ditR* mutant (formerly Δ*smlt0547*) harboring pP$_{mfsBC-lacZ}$, and the *mfsBC* promoter activity in response to diamide was determined. The results in Fig 5 illustrate that the replacement of either Cys92 or Cys127 diminished the diamide-inducible property of DitR. This finding strongly suggests the involvement of Cys92 and Cys127 in the diamide sensing mechanism of *S. maltophilia* DitR. The inactivation of the TetR-type transcriptional repressor by cysteine modification has been reported in *E. coli*, where NemR, formerly called YdhM, regulates *gloA* and *nemA* encoding enzymes involved in the degradation of α-ketoaldehydes, electrophiles, and thiol-depleting agent NEM [33]. NemR likely senses and responds to NEM and alkylating agents through thiol-based oxidation of cysteine residues [33, 34]. Although the inactivation of both NemR and DitR repressors is mediated by the modification of cysteine residues, the findings that DitR shares a very low identity (15.9%) with NemR and NEM is not a potent inducer of DitR suggest the different mechanisms through which NemR and DitR are used to sense thiol-depleting agents.

## Characterization of *mfsBC* promoter

The putative *mfsBC* promoters were localized by determining the transcriptional start site (+1). The 5′ RACE was performed to determine the 5′ end of the *mfsBC* transcript. The RACE-PCR products were cloned into the pUC18 vector and sequenced. Most sequences from four out of the seven independent clones mapped the 5′ end of the *mfsBC* transcript to A residue, located 65 nucleotides upstream of the translational start codon (ATG). The *mfsBC*

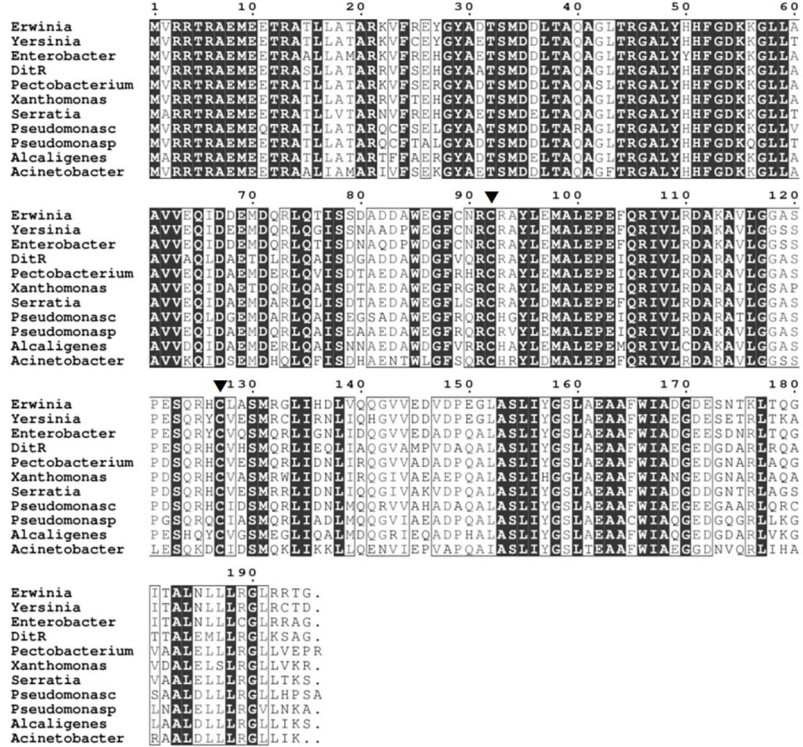

**Fig 8. Multiple alignment of *S. maltophilia* DitR with other orthologs.** Amino acid alignment was performed using Clustal omega. Arrowheads indicate the conserved cysteine residues, Cys92 and Cys127. Erwinia, *Erwinia persicina*: CI789_09880; Yersinia, *Yersinia canariae*: F0T03_09235; Enterobacter, *Enterobacter sp*. 638: Ent638_2692; Pectobacterium, *Pectobacterium carotovorum* subsp. carotovorum PCC21: PCC21_025910; Xanthomonas, *Xanthomonas arboricola*: XB05_05075; Serratia, *Serratia fonticola* GS2: AV650_21160; Pseudomonasc. *Pseudomonas cremoricolorata*: LK03_16165; Pseudomonasp, *Pseudomonas putida* GB-1: PputGB1_2508; Alcaligenes, *Alcaligenes faecalis* JQ135: AFA_03875; Acinetobacter, *Acinetobacter bereziniae*: BSR55_08870.

promoter had sequence motifs GATTAT and TGTAGG corresponding to putative -35 and -10 regions, respectively, which were separated by 20 nucleotides (Fig 9A).

## Localization of DitR binding sites on putative *mfsBC* promoters

DNase I footprinting assays were performed to localize the binding position of DitR protein on the *mfsBC* promoter sequence. The purified DitR protein and the [32]P-labeled *mfsBC* promoter fragment were mixed in the binding reaction prior to DNase I digestion. Fig 9B shows two sites of DNase I protection: Box I and Box II. Box I sequence (5′CATACGCTCTGTATG‐TATTGTC3′), which contains an inverted repeat CATACnnnnnGTATG, spans from position +21 to -1 corresponding to the *mfsBC* promoter, thereby covering the +1 site (Fig 9B). Box II has an extra-long sequence 5′TACATACGCAGTGCATGATTATTTACATACGACGCGTAT‐GAATGTAG3′ starting from position -11 to -57. When DitR binding Box I was aligned with Box II, two sequences that share high identity with Box I (i.e., 5′CATACGCAGTGCATGAT‐TATTT3′ and 5′CATACGACGCGTATGAATGTAG3′) were identified and named Box IIa and Box IIb, respectively. The inverted repeat CATACnnnnnGTATG was conserved for both Box IIa and Box IIb, suggesting that DitR binds to the box as a homodimer similar to a typical TetR transcriptional repressor [35]. The DitR binding Box IIb is located between -35 and -10 motifs (at position -32 to -11) of the *mfsBC* promoter (Fig 9A), while Box IIa is located immediately upstream of the -35 motif at positions -55 to -34. It is likely that the *mfsBC* promoter

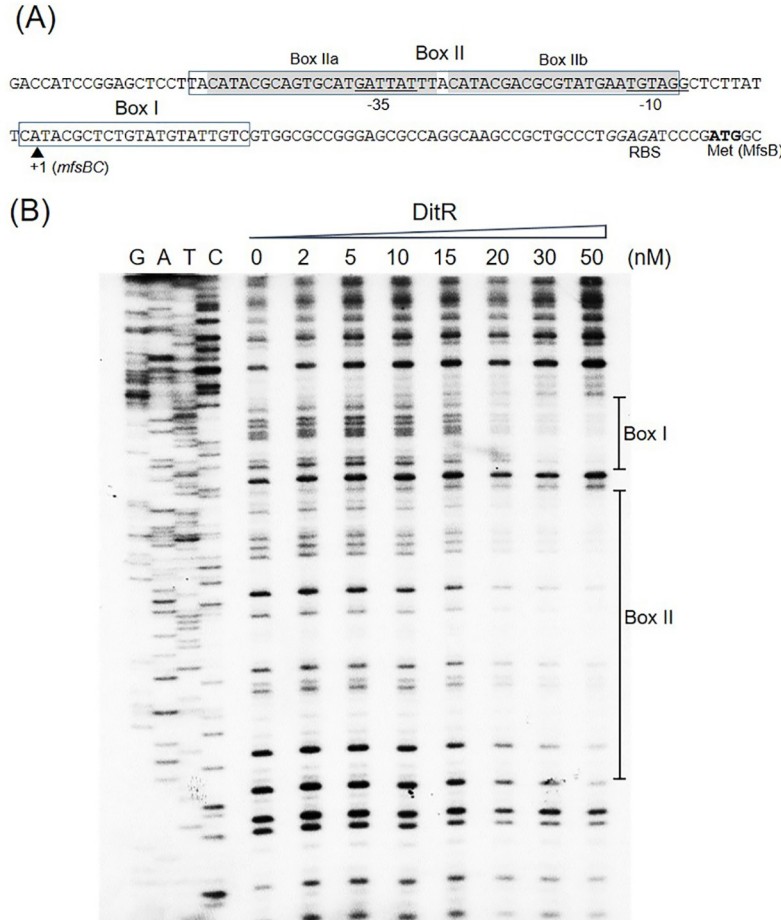

**Fig 9. Characterization of the *mfsBC* promoter and localization of DitR.** (A) Putative promoter region of the *msfBC* operon was characterized. The translation start codon (ATG) and putative ribosome binding site (RBS) are indicated in bold and italic fonts, respectively. Arrowhead indicates putative transcription start site (+1), and the -35 and -10 motifs are underlined. DitR-binding Box I and Box II are boxed, and Box IIa and Box IIb are shaded in gray. (B) DNase I footprinting assay was conducted using increasing concentrations of purified DitR (0–50 nM) and $^{32}$P-labeled 204-bp *mfsBC* promoter sequence. G, A, T, C is the sequence ladder.

contains three DitR binding sites. This was supported by the EMSA result, where multiple band shifts were detected (Fig 7A). The binding of DitR to these binding sites would hinder the binding of RNA polymerase to the *mfsBC* promoter, thereby inhibiting the transcription of *mfsB* and *mfsC*.

## Conclusion

We showed here the role of the MfsC transporter in protecting *S. maltophilia* from the toxicity of diamide. *mfsC* was transcribed as a polycistronic mRNA with *mfsB*, which encodes MFS with an unknown function. The expression of *mfsBC* is controlled by the DitR repressor. Under physiological conditions, DitR binds three binding boxes spanning the *mfsBC* promoter sequence and represses the transcription of the *mfsBC* operon. Upon exposure to diamide, the oxidation of DitR at Cys92 and Cys127 leads to a release of DitR from the *mfsBC* promoter and, therefore, enables the binding of RNA polymerase to initiate *mfsBC* transcription. This adaptive response would help *S. maltophilia* survive diamide stress.

## Supporting information

**S1 Fig. Purity of purified DitR protein.** The coomassie blue staining of DitR proteins (2, 5 and 10 μg) after purification and separation by 12.5% SDS-PAGE. M represents protein molecular weight markers.
(TIF)

**S2 Fig. Uncropped and unmodified image of end-point RT-PCR (Fig 4B).**
(TIF)

**S3 Fig. Uncropped and unmodified image of gel mobility shift assay (Fig 7A).**
(TIF)

**S4 Fig. Uncropped and unmodified image of gel mobility shift assay (X-ray film) (Fig 7A).**
(TIF)

**S5 Fig. Uncropped and unmodified image of gel mobility shift assay (Fig 7B).**
(TIF)

**S6 Fig. Uncropped and unmodified image of gel mobility shift assay (X-ray film) (Fig 7B).**
(TIF)

**S7 Fig. Uncropped and unmodified image of DNase I footprinting assay (X-ray film) (Fig 9B).**
(TIF)

**S8 Fig. Uncropped and unmodified image of DNase I footprinting assay (scanned X-ray film) (Fig 9B).**
(TIF)

**S9 Fig. Uncropped and unmodified image of SDS-PAGE of S1 Fig.**
(TIF)

**S1 Table. Raw data for antibiotic susceptibility test of Δ*mfsB* and Δ*mfsC* mutants, compared to wild type strain (Fig 2A).**
(XLSX)

**S2 Table. Raw data for determination of survival (%) of Δ*mfsB* and Δ*mfsC* mutants, and wild type strain against oxidants (Fig 2B).**
(XLSX)

**S3 Table. Raw data for investigation of *mfsC* transcriptional level in the presence of several oxidants (Fig 3).**
(XLSX)

**S4 Table. Raw data for *in vivo* characterization of site-directed mutagenesis of DitR on *mfsBC* promoter (Fig 5).**
(XLSX)

**S1 Raw images.**
(PDF)

## Acknowledgments

The authors thank Weerachai Tanboon for technical assistance. Parts of this work are from AB dissertation submitted for a Ph. D. degree from Mahidol University.

## Author Contributions

**Conceptualization:** Paiboon Vattanaviboon.

**Data curation:** Angkana Boonyakanog, Nisanart Charoenlap.

**Formal analysis:** Nisanart Charoenlap.

**Funding acquisition:** Paiboon Vattanaviboon.

**Investigation:** Angkana Boonyakanog, Sorayut Chattrakarn.

**Methodology:** Angkana Boonyakanog, Sorayut Chattrakarn.

**Resources:** Skorn Mongkolsuk.

**Supervision:** Nisanart Charoenlap, Paiboon Vattanaviboon, Skorn Mongkolsuk.

**Validation:** Paiboon Vattanaviboon.

**Visualization:** Paiboon Vattanaviboon.

**Writing – original draft:** Angkana Boonyakanog, Paiboon Vattanaviboon.

**Writing – review & editing:** Nisanart Charoenlap, Paiboon Vattanaviboon, Skorn Mongkolsuk.

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
