## [Decision Letter · Decision Letter 0]

31 May 2022

PONE-D-22-09616Contribution of Stenotrophomonas maltophilia MfsC transporter to protection against diamide and the regulation of its expression by the diamide responsive repressor DitRPLOS ONE

Dear Dr. Vattanaviboon,

Thank you for submitting your manuscript to PLOS ONE. After careful consideration, we feel that it has merit but does not fully meet PLOS ONE’s publication criteria as it currently stands. Therefore, we invite you to submit a revised version of the manuscript that addresses the points raised during the review process.

We look forward to receiving your revised manuscript.

Kind regards,

Yixin Shi

Academic Editor

PLOS ONE

Journal Requirements:

"This work was supported by Thailand Science Research and Innovation (TSRI), Chulabhorn Research Institute (Grant number 313/2231) and Chulabhorn Graduate Institute (Grant number 2563/007). AB was a recipient of the Science Achievement Scholarship of Thailand from the Office of Higher Education Commission of the Government of Thailand."

We note that you have provided funding information. However, funding information should not appear in the Funding section or other areas of your manuscript. We will only publish funding information present in the Funding Statement section of the online submission form. 

"This work was supported by Thailand Science Research and Innovation (TSRI), Chulabhorn Research Institute (Grant number 313/2231) and Chulabhorn Graduate Institute (Grant number 2563/007). AB was a recipient of the Science Achievement Scholarship of Thailand from the Office of Higher Education Commission of the Government of Thailand.  The funders had no role in study design, data collection and analysis, decision to publish, or preparation of the manuscript."

Reviewers' comments:

Reviewer's Responses to Questions

**Comments to the Author**

1. Is the manuscript technically sound, and do the data support the conclusions?

Reviewer #1: Yes

Reviewer #2: Yes

2. Has the statistical analysis been performed appropriately and rigorously? 

Reviewer #1: Yes

Reviewer #2: No

3. Have the authors made all data underlying the findings in their manuscript fully available?

Reviewer #1: Yes

Reviewer #2: Yes

4. Is the manuscript presented in an intelligible fashion and written in standard English?

Reviewer #1: Yes

Reviewer #2: Yes

5. Review Comments to the Author

Reviewer #1: The manuscript presented characterization of MfsC transporter to protection against diamide and the regulation of its expression by the diamide responsive repressor DitR in Stenotrophomonas maltophilia. Before the paper could be accepted for publication in PLOS ONE, the authors need make revisions as follow:

1. What’s the function of MfsB? What’s the relationship between MfsB and MfsC? Please add discussion.

Reviewer #2: This is a very nice characterization of a transporter operon and its regulator that mediated resistance to diamide in Stenotrophomonas maltophilia (Steno). The authors show deletion of mfsC leads to increased sensitivity to diamide and that the regulator smlt0547 (named ditR here) is a repressor (based on deletion, complementation, and EMSA behavior) that binds the mfsC operon promoter (shown by EMSA and DNase footprinting) and is released by the presence of diamide (shown by EMSA) resulting in operon expression specific to diamide presence (qRT-PCR and reporter assays). Release by diamide is reversible by DTT treatment and loss of conserved reactive cysteines in DitR result in a diamide-insensitive and thus constitutively repressive protein (shown by reporter expression). These findings are related to the field via appropriate discussion and phylogenetic analyses. I think this is a really nice story and really enjoyed reading it. I have a few minor things to suggest or to be clarified, followed by some suggested text edits to make the manuscript more clear.

1. For the various compounds, particularly the oxidants, why were these concentrations chosen? It seems as if diamide is used at the highest concentration and one could argue that the repressor could be responsive and/or transporter active if the other compounds were present at higher concentrations. I do not think this is the case, but an explanation of why these concentrations were chosen would be very helpful.

2. Statistics: For all of the comparisons in which there are more than two groups (all of them) any t-tests need to be corrected for multiple comparisons by an appropriate post-test. For Fig 3 it would be a Dunnett’s post-test with WT as the comparator tested within each compound. For Fig 4 it would be a Dunnett’s post-test with untreated as the comparator. I don’t think this increased statistical rigor will change the conclusions but use of multiple-comparison corrected stats is important.

3. Figure suggestions: Fig 1 is unnecessary, the description in the text is sufficient. For the phylogenetic trees, I would prefer plotting the trees with quantitative branch lengths (actual branch lengths with an amino-acid difference scale) so that relative similarity is visually represented.

4. Methods: Line 97: I find it really hard to believe that a bunch of molecular biology was conducted according to a 21 year old method manual without the use of any kits or any deviations from those published methods. Please expand detail as needed. Line 110: Which plasmid was the cre recombinase expressed from and how was the strain cured of that plasmid?

5. This is really just a comment to improve future work. Having myself tried lacZ in Steno many times, it really is not a great reporter and seems to be due to poor protein folding. The authors have appropriately used a Unit based measure, as these numbers would look ridiculously low if the common Miller unit were used (1 Bgal unit in this manuscript would be 0.002 Miller Units, if my calculations are correct). I don’t think anything needs to change for this manuscript, but I would suggest using a better reporter for Steno in the future. XylE works very well as an enzymatic reporter and mScarlet works really well as a fluorescent reporter.

Text edit suggestions:

Line 25: change ‘transports’ to ‘transporters’

Line 44: edit first sentence to- “Stenotrophomonas maltophilia is a Gram-negative, rod-shaped, aerobic bacterium, which is ubiquitous in the aqueous environment and soil.”

Line 50: edit start of sentence to- “Efflux transporters are recognized…”

Line 52: Since the assumption of kingdom number keeps changing, I suggest using “all domains of life” instead of numbering the kingdoms.

Line 57: change ‘several’ to ‘many’ and replace ‘however’ with ‘and’

Line 82: suggest “Here, we are interested in the MFS transporters because of their…”

Line 83: change ‘gene’ to ‘transcriptional’

Line 86: Suggest putting the smlt gene number for DitR and stating ‘… which we have named DitR’

Line 351: change ‘direction’ to ‘orientation’

6. PLOS authors have the option to publish the peer review history of their article (what does this mean?). If published, this will include your full peer review and any attached files.

Reviewer #1: No

Reviewer #2: No

---

## [Author Response · Author response to Decision Letter 0]

21 Jun 2022

Point-by-point Response to Reviewers

PONE-D-22-09616

Contribution of Stenotrophomonas maltophilia MfsC transporter to protection against diamide and the regulation of its expression by the diamide responsive repressor DitR

PLOS ONE

Reviewer #1: The manuscript presented characterization of MfsC transporter to protection against diamide and the regulation of its expression by the diamide responsive repressor DitR in Stenotrophomonas maltophilia. Before the paper could be accepted for publication in PLOS ONE, the authors need make revisions as follow:

1. What’s the function of MfsB? What’s the relationship between MfsB and MfsC? Please add discussion.

Reply: According to our current data, the function of MfsB is still unknown. The �mfsB mutant showed no altered phenotypes against all tested oxidants, antibiotics and biocides. mfsB was co-transcribed with mfsC and its expression could be induced by diamide. We have added discussion about this mfsB as “Although mfsB was co-transcribed with mfsC in an diamide-inducible manner, it seems likely that MfsB has no role in the protection of S. maltophilia against all tested antibiotics and oxidants (Fig 2). The precise physiological role of S. maltophilia MfsB is as yet unknown and required further investigation.” (line 344-347)

Reviewer #2: This is a very nice characterization of a transporter operon and its regulator that mediated resistance to diamide in Stenotrophomonas maltophilia (Steno). The authors show deletion of mfsC leads to increased sensitivity to diamide and that the regulator smlt0547 (named ditR here) is a repressor (based on deletion, complementation, and EMSA behavior) that binds the mfsC operon promoter (shown by EMSA and DNase footprinting) and is released by the presence of diamide (shown by EMSA) resulting in operon expression specific to diamide presence (qRT-PCR and reporter assays). Release by diamide is reversible by DTT treatment and loss of conserved reactive cysteines in DitR result in a diamide-insensitive and thus constitutively repressive protein (shown by reporter expression). These findings are related to the field via appropriate discussion and phylogenetic analyses. I think this is a really nice story and really enjoyed reading it. I have a few minor things to suggest or to be clarified, followed by some suggested text edits to make the manuscript more clear.

1. For the various compounds, particularly the oxidants, why were these concentrations chosen? It seems as if diamide is used at the highest concentration and one could argue that the repressor could be responsive and/or transporter active if the other compounds were present at higher concentrations. I do not think this is the case, but an explanation of why these concentrations were chosen would be very helpful. 

Reply: The oxidant concentrations that we used in expression analysis are the sub-minimal inhibitory concentrations (sub-MIC). We have added this statement on line 187. 

2. Statistics: For all of the comparisons in which there are more than two groups (all of them) any t-tests need to be corrected for multiple comparisons by an appropriate post-test. For Fig 3 it would be a Dunnett’s post-test with WT as the comparator tested within each compound. For Fig 4 it would be a Dunnett’s post-test with untreated as the comparator. I don’t think this increased statistical rigor will change the conclusions but use of multiple-comparison corrected stats is important.

Reply: Thank you for this valuable suggestion. We have changed statistical analysis to Dunnett’s post-test as suggestions and we found that it did not affect conclusions in both Fig 3 and Fig 4.

3. Figure suggestions: Fig 1 is unnecessary, the description in the text is sufficient. For the phylogenetic trees, I would prefer plotting the trees with quantitative branch lengths (actual branch lengths with an amino-acid difference scale) so that relative similarity is visually represented.

Reply: Fig 1 has been removed as suggestion. The branch length of the phylogenetic trees in Fig. 2 (now Fig 1) and Fig 7 (now Fig 6) has been changed to an actual branch length to visually present the differences as suggestion.

4. Methods: Line 97: I find it really hard to believe that a bunch of molecular biology was conducted according to a 21 year old method manual without the use of any kits or any deviations from those published methods. Please expand detail as needed. Line 110: Which plasmid was the cre recombinase expressed from and how was the strain cured of that plasmid?

Reply: Thank you for your comments. We did use some kits for plasmid preparation and extraction of DNA fragments from agarose gel. We have mentioned this on line 100-102. 

The Cre-recombinase protein was expressed from pCM157 plasmid. We have added more detail about the excision of Gmr cassette and curing of pCM157 as “To excise the Gmr marker, pCM157 [15] expressing Cre-recombinase was introduced into the putative mutant and the resultant unmarked ΔmfsB mutants obtained from Cre-mediated loxP recombination were screened for the gentamicin sensitive phenotype. The plasmid pCM157 (carrying tetracycline-resistant gene) was cured by serial subcultured the strains on plates without antibiotic supplementation and screened for the tetracycline-sensitive colonies.” (line 111-116).

5. This is really just a comment to improve future work. Having myself tried lacZ in Steno many times, it really is not a great reporter and seems to be due to poor protein folding. The authors have appropriately used a Unit based measure, as these numbers would look ridiculously low if the common Miller unit were used (1 Bgal unit in this manuscript would be 0.002 Miller Units, if my calculations are correct). I don’t think anything needs to change for this manuscript, but I would suggest using a better reporter for Steno in the future. XylE works very well as an enzymatic reporter and mScarlet works really well as a fluorescent reporter.

Reply: We appreciate your valuable suggestions and it may be useful for our future works. 

6. Text edit suggestions:

Line 25: change ‘transports’ to ‘transporters’

Reply: We have edited as suggestion.

Line 44: edit first sentence to- “Stenotrophomonas maltophilia is a Gram-negative, rod-shaped, aerobic bacterium, which is ubiquitous in the aqueous environment and soil.”

Reply: We have edited as suggestion.

Line 50: edit start of sentence to- “Efflux transporters are recognized…”

Reply: We have edited as suggestion.

Line 52: Since the assumption of kingdom number keeps changing, I suggest using “all domains of life” instead of numbering the kingdoms.

Reply: We have edited as suggestion.

Line 57: change ‘several’ to ‘many’ and replace ‘however’ with ‘and’

Reply: We have edited as suggestion.

Line 82: suggest “Here, we are interested in the MFS transporters because of their…”

Reply: We have edited as suggestion.

Line 83: change ‘gene’ to ‘transcriptional’

Reply: We have edited as suggestion.

Line 86: Suggest putting the smlt gene number for DitR and stating ‘… which we have named DitR’

Reply: We have edited as suggestion.

Line 351: change ‘direction’ to ‘orientation’

Reply: We have edited as suggestion.

---

## [Decision Letter · Decision Letter 1]

19 Jul 2022

Contribution of Stenotrophomonas maltophilia MfsC transporter to protection against diamide and the regulation of its expression by the diamide responsive repressor DitR

PONE-D-22-09616R1

Dear Dr. Vattanaviboon,

We’re pleased to inform you that your manuscript has been judged scientifically suitable for publication and will be formally accepted for publication once it meets all outstanding technical requirements.

Kind regards,

Yixin Shi

Academic Editor

PLOS ONE

Reviewers' comments:

Reviewer's Responses to Questions

**Comments to the Author**

1. If the authors have adequately addressed your comments raised in a previous round of review and you feel that this manuscript is now acceptable for publication, you may indicate that here to bypass the “Comments to the Author” section, enter your conflict of interest statement in the “Confidential to Editor” section, and submit your "Accept" recommendation.

Reviewer #1: (No Response)

Reviewer #2: All comments have been addressed

2. Is the manuscript technically sound, and do the data support the conclusions?

Reviewer #1: Yes

Reviewer #2: Yes

3. Has the statistical analysis been performed appropriately and rigorously? 

Reviewer #1: Yes

Reviewer #2: Yes

4. Have the authors made all data underlying the findings in their manuscript fully available?

Reviewer #1: Yes

Reviewer #2: Yes

5. Is the manuscript presented in an intelligible fashion and written in standard English?

Reviewer #1: Yes

Reviewer #2: Yes

6. Review Comments to the Author

Reviewer #1: The responses to reviewer’s comments were clear, and the revised manuscript was satisfied. The authors know the importance of elucidation of MfsB function and the relationship between MfsB and MfsC. I recommend the paper could be accepted for publication in PLOS ONE.

Reviewer #2: (No Response)

7. PLOS authors have the option to publish the peer review history of their article (what does this mean?). If published, this will include your full peer review and any attached files.

Reviewer #1: No

Reviewer #2: No

---

## [Editor Report · Acceptance letter]

22 Jul 2022

PONE-D-22-09616R1 

Contribution of *Stenotrophomonas maltophilia* MfsC transporter to protection against diamide and the regulation of its expression by the diamide responsive repressor DitR 

Dear Dr. Vattanaviboon:

I'm pleased to inform you that your manuscript has been deemed suitable for publication in PLOS ONE. Congratulations! Your manuscript is now with our production department. 

Kind regards, 

on behalf of

Dr. Yixin Shi 

Academic Editor

PLOS ONE